# Research on Lightning Overvoltage Protection of Line-Adjacent Pipelines Based on Solid-State Decoupling

Wei Liu [1], Yuanchao Hu [2,*], Haipeng Tian [2], Zhipeng Jiang [1], Xiaole Su [1], Jie Xiong [1], Wei Su [1] and Yi Wang [1]

[1] State Grid Hubei Power Supply Limited Company Ezhou Power Supply Company, Ezhou 436000, China; ezliuwei@sina.com (W.L.); ezjzp_80@sina.com (Z.J.); ezsuxiaole@outlook.com (X.S.); matix90@yeah.net (J.X.); du990814@163.com (W.S.); yc2023126@126.com (Y.W.)

[2] School of Electrical and Electronic Engineering, Shandong University of Technology, Zibo 255000, China; tianhaipeng1999@163.com

* Correspondence: huyuanchao3211@126.com

**Abstract:** Existing transmission lines and pipelines are frequently crossed and erected in parallel, meaning that if lightning strikes a wire and causes insulator flashovers, the resulting lightning current will spread through the grounding of the tower where the flashover insulator is located. This dispersion of current can lead to overvoltage effects on nearby pipelines. This study performs simulation calculations to analyze the overvoltage experienced by pipelines due to the dispersion of grounding current from the tower. Furthermore, this paper proposes a method for protecting the pipeline from such an overvoltage. Firstly, the lightning transient calculation model of a transmission line tower is constructed using the electromagnetic transient software ATP-EMTP 5.5. The model calculates the effects of lightning peak currents and soil resistivity on the distribution characteristics of lightning current in the tower, specifically in the area where the flashover insulator is located. Subsequently, a calculation model of the tower grounding grid–natural gas pipeline is developed, taking into account the distribution characteristics of lightning current in the tower. This model analyzes the impact of lightning peak currents, soil resistivity, and pipeline spacing on pipeline overvoltage. Finally, the effectiveness of the solid-state decoupler in mitigating lightning overvoltage in the pipeline is verified. The results demonstrate a positive correlation between the lightning current entering the tower grounding grid through the flashover insulator and the lightning current distribution characteristics. The solid-state decoupling device proves to be effective in reducing the voltage of the pipeline insulation layer, and the simulation results provide the optimal laying length of the bare copper wire.

**Keywords:** lightning stroke conductor; natural gas pipeline; grounding current divergence; pipeline overvoltage protection

## 1. Introduction

Since the national "new infrastructure" and other development strategies were put forward, in which the "energy Internet" has become an important form of construction, the "West–East Gas Transmission", "West-to-East Electricity Transmission", and other national cross-regional energy transmission projects have been implemented, effectively alleviating China's energy load center distribution imbalance and other issues to a great extent, to meet the social development needs of electricity, oil and natural gas, and other energy forms [1]. However, the new infrastructure also faces many problems, for example, to save land area, electricity, oil and gas, and other energy supply facilities often use the "shared corridor" construction method, leading to the inevitable formation of natural gas pipelines, electric power overhead line crossovers, and so on, which cause "two lines and one land" electromagnetic interference problems that are gradually becoming larger concerns [2,3]. When the overhead power conductor suffers a lightning strike, the lightning current in the intruding conductor will generate a space-alternating electromagnetic field

around the conductor such that the adjacent pipeline, due to inductive coupling, produces a pipeline overvoltage [4,5]. If the lightning current makes the line insulator flashover, the lightning current will be scattered along the surface of the flashover insulator through the tower grounding grid such that the adjacent pipeline, due to resistive coupling, produces a pipeline overvoltage [6–8]. Furthermore, due to the existence of resistive coupling and the pipeline insulation layer's potential difference, the safe operation of the pipeline has been threatened; thus, research into the overvoltage protection of the transmission line adjacent to the pipeline is of great significance.

Protection for the transmission line near the natural gas pipeline mainly includes two aspects: the first concerns protection for the DC transmission line, research into the DC leakage current near the natural gas pipeline, and metal layer corrosion research to determine some measures to slow down or inhibit the metal layer corrosion; the second concerns the normal operation of the overhead line, research into grounding faults, and how a lightning strike near the natural gas pipeline induces overvoltage through studying some measures to reduce the magnitude of the pipeline overvoltage.

Lucca Giovanni studied the effect of pipe insulation coating breakage on electromagnetic interference between transmission lines and pipelines and proposed a law stating that the resistance of pipe insulation to soil decreases over time [9]. Mo Bingyu et al. used Maxwell's system of equations to derive the pipeline-induced electromotive force originating from the AC transmission line; established the pipeline equivalent circuit; analyzed the electromagnetic coupling to the neighboring underground pipelines during the steady-state operation of the AC transmission line; and, combined with a practical case, proposed a measurement method that can be corrected to the pipeline–ground potential distribution at the same moment [10]. Based on the influence of the pipeline coating resistance on the pipeline characteristic impedance, M. Rabbani concluded that the smaller the impedance, the smaller the induced voltage on the pipeline; conversely, the greater the impedance of the coating, the greater the voltage difference between the pipeline and the coating [11]. G.T. Quickel, by characterizing accidentally perforated pipes, concluded that the cause of the pipe perforation is a high-voltage arcing discharge from the pipe wall to the ground, which is mainly due to lightning strikes or the occurrence of ground faults in the overhead lines, and proposed three methods for protecting the insulation joint of the pipeline [12].

A significant amount of research has been conducted both domestically and internationally on protecting natural gas pipelines from electromagnetic interference caused by overhead lines. This research primarily focuses on two aspects: steady-state interference protection and lightning overvoltage protection. Currently, solid-state decouplers are commonly used for steady-state interference protection, while limited research has been performed on lightning overvoltage protection for pipelines. This paper aims to investigate the impact of grounded, scattered lightning currents on nearby natural gas pipelines and assess the effectiveness of solid-state decouplers in reducing the voltage endured by the pipeline's insulating layer. The simulation results are then used to determine the optimal length for laying bare copper wire, as presented in the provided equation.

## 2. Simulation Model and Parameters

### 2.1. Line Tower Model and Parameters

In order to study the impact of lightning strikes more accurately on overhead conductors regarding the overvoltage of natural gas pipelines in the vicinity, it is essential to calculate the distribution of lightning current between overhead lines and towers after a lightning strike. For this study, a 500 kV overhead line is selected and the frequency-dependent JMarti line model in ATP-EMTP is utilized to determine the line parameters [13,14]. The relevant parameters for the overhead conductor and lightning line can be found in Table 1. The line insulators consist of 28 glass insulators of type U210BP/170, and voltage-controlled switches are employed to model these insulators in ATP-EMTP [15,16].

**Table 1.** Parameters of conductors and lightning conductors.

| Model Numbers | Outside Diameter/mm | DC Resistance/($\Omega \cdot km^{-1}$) | Fission Number |
|---|---|---|---|
| LGJ-400/35 | 26.82 | 0.07389 | 4 |
| JLB20A-80 | 11.4 | 1.0788 | 0 |

The tower mode chosen for this study is a more realistic equidistributed, multi-wave impedance tower model, and the tower is taken to be a ZB4 wine-glass-type linear tower. The structure of the power pole tower is illustrated in Figure 1.

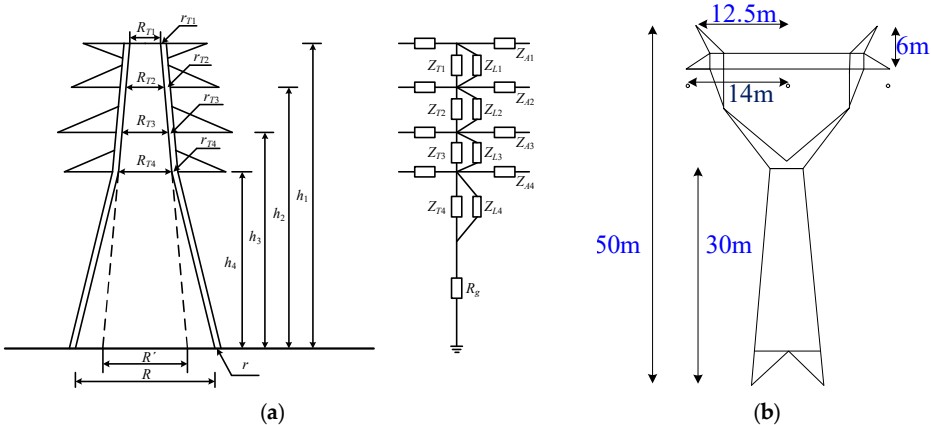

(a)           (b)

**Figure 1.** The tower model and parameters used in ATP-EMTP software are calculated. (**a**) Tower model. (**b**) Electric power tower structure.

### 2.2. Modeling and Parameterization of Tower Grounding Grids and Natural Gas Pipelines

In the actual project of power overhead line tower grounding grid structure, various types are used. The most commonly used type is the box with a ray type grounding grid. For this study, a typical grounding grid is selected, with a diameter of $\Phi = 16$ mm and a buried depth of $h_1 = 0.8$ m, using galvanized round steel as the material. As shown in Figure 2, the length of grounding grid box is 15 m and the length of the extended ray is 20 m. The angle between the extension ray and the box extension line is $\alpha = 45°$. When calculating the distribution of lightning current, it is sufficient to model it as a grounding resistance.

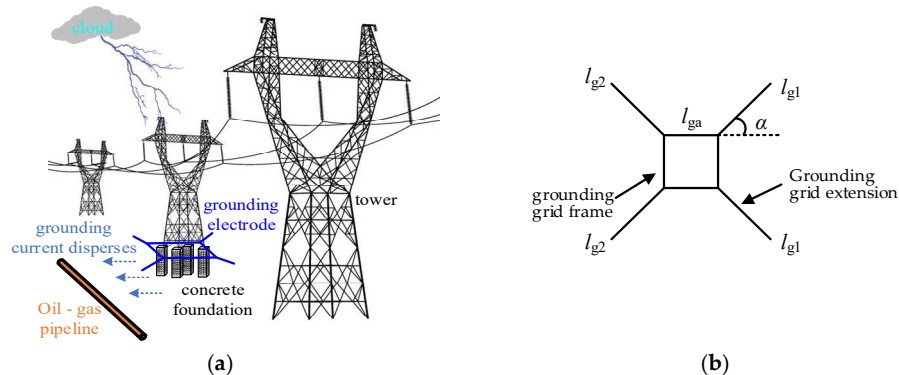

(a)           (b)

**Figure 2.** Tower grounding grid structure. (**a**) Lightning current dispersion path. (**b**) Tower grounding grid structure.

The selection of the natural gas pipeline is based on a standard, exemplified by the cross-section of the pipeline shown in Figure 3. The pipe insulation layer material used is 3PE, with the specific parameters provided in Table 2.

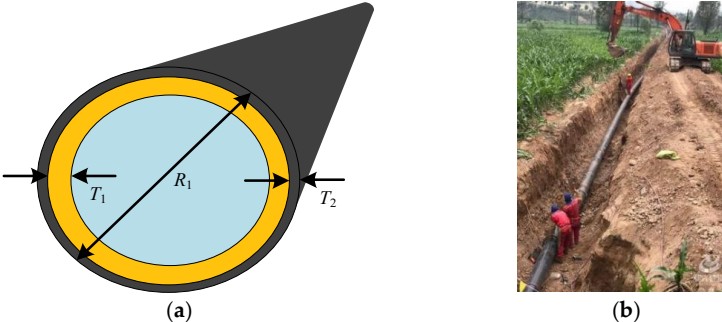

(**a**)                                    (**b**)

**Figure 3.** Natural gas pipeline profile and construction site. (**a**) Natural gas pipeline section diagram. (**b**) Natural gas pipeline construction site.

**Table 2.** Pipeline parameters.

| Parameter Name | Numerical Value |
|---|---|
| Outside diameter/mm | 429 |
| Steel thickness/mm | 9.5 |
| Insulation thickness/mm | 3 |

To investigate the effects of overvoltage on a natural gas pipeline at a cross-crossing after a lightning strike on the conductor, causing flashover of the line insulator and resulting in ground scattering current, a simplified model of the $T_0$, $T_{R1}$ tower grounding grid and the natural gas pipeline (as depicted in Figure 4b) is established using COMSOL Multiphysics 5.6 simulation software. The length of the natural gas pipeline ($l_p$) is 400 m, and both sides of the pipeline and the $T_0$, $T_{R1}$ tower grounding grid have a pendant angle of (β). Both sides of the pipeline are treated with grounding, and the burial depth (h2) is 2 m.

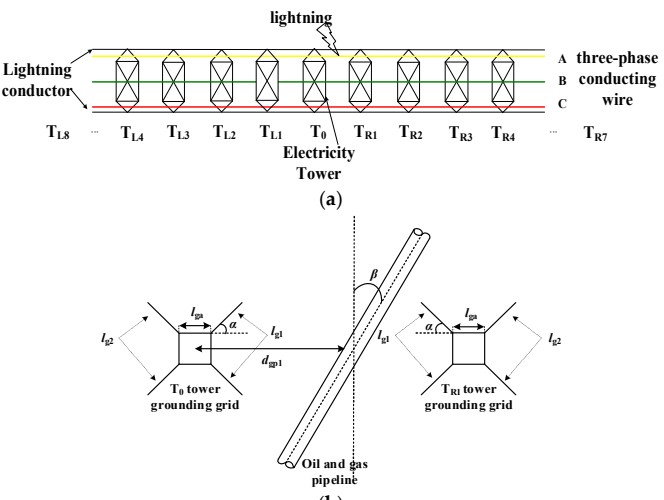

(**a**)

(**b**)

**Figure 4.** Simulation diagram of tower grounding grid and pipeline. (**a**) Schematic diagram of lightning conductor. (**b**) Top view of tower grounding grid and natural gas pipeline.

## 3. Calculation of Lightning Current Distribution Characteristics

### 3.1. The Influence of Lightning Peak Current

ATP-EMTP is utilized to establish the transmission tower line model. Figure 4a depicts its relative position. The lightning current is modeled using a double exponential model with a waveform parameter of 2.6/50 μs. The lightning stroke point occurs at the A-phase conductor in the center of the $T_0$ and $T_{R1}$ spans. With a soil resistivity of ρ = 1000 Ω m, the simulation results indicate that the lightning withstand level of shielding failure for a

500 kV overhead line is 24.18 kA. It is observed that when the amplitude of the lightning current is below 24.18 kA, the majority of the lightning current flows through the overhead conductor. Keeping other simulation parameters constant, the lightning peak current ($I_0$) is varied to 5 kA, 30 kA, 35 kA, 40 kA, 45 kA, and 50 kA to assess its impact on the lightning current distribution after line insulator flashover. Figure 5 illustrates the lightning current distribution at the $T_0$ and $T_{R1}$ towers.

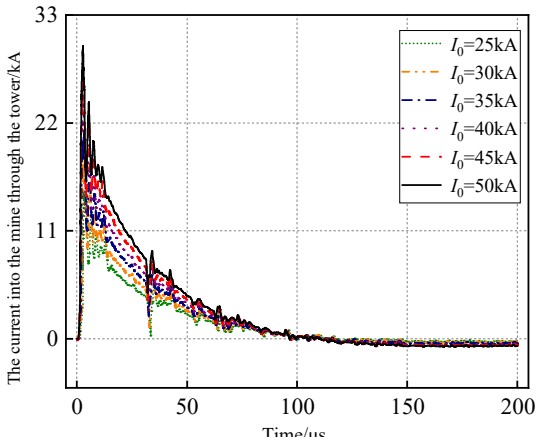

**Figure 5.** The influence of lightning peak current on the lightning current distribution of $T_0$ and $T_{R1}$ towers.

According to the results shown in Figure 5, it is observed that when the amplitude of the lightning current exceeds 24.18 kA, the line insulators of T0 and TR1 towers experience flashover. This occurs because the lightning strike point is located at the center of the overhead wire span between $T_0$ and $T_{R1}$ towers. As a result, the lightning current is dispersed into the ground through the $T_0$ and $T_{R1}$ tower grounding grid. Moreover, as the amplitude of the lightning current increases, the current flowing through the $T_0$ and $T_{R1}$ tower grounding grid also gradually increases. Consequently, the overvoltage issue of the $T_0$ and $T_{R1}$ tower grounding current affecting the adjacent pipeline needs to be taken into consideration.

### 3.2. The Influence of Soil Resistivity

The distribution of lightning current after the flashover of line insulators is studied by varying the soil resistivity ρ. The lightning peak current $I_0$ is fixed at 50 kA. The different values of soil resistivity considered are 200 Ω m, 500 Ω m, 800 Ω m, 1000 Ω m, 1200 Ω m, and 1500 Ω m. Figure 6 illustrates the lightning current distribution for $T_0$ and $T_{R1}$ towers.

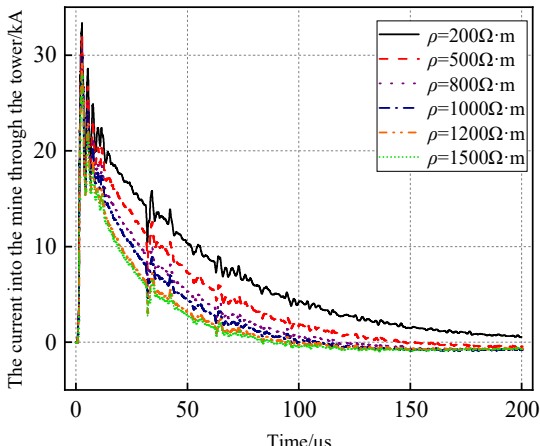

**Figure 6.** The influence of soil resistivity on lightning current distribution of T0 and TR1 towers.

The results indicate that as soil resistivity increases, the mine current entering the tower through $T_0$ and $T_{R1}$ decreases. Additionally, the waveform and amplitude of the mine current entering the tower through $T_0$ and $T_{R1}$ remain the same. This phenomenon occurs because, following a line insulator flashover, the increase in soil resistivity leads to an increase in the grounding resistance of the tower grounding grid. This increase in resistance partially obstructs the flow of lightning current into the ground through the tower grounding grid. Simultaneously, the increase in soil resistivity also hampers the flow of lightning current through the tower grounding grid into the surrounding soil area.

## 4. Analysis of Pipeline Overvoltage under the Action of Multi-Tower Flow

When the power overhead conductor is struck by lightning, the lightning current can cause the line insulator to flashover. In this situation, the lightning current disperses along the surface of the flashover insulator through the tower grounding grid. Due to the presence of resistive coupling, the potential of the soil area near the grounding grid and the steel layer of the adjacent pipeline will increase. As a result, there will be a potential difference between the soil area near the pipeline and the steel layer of the pipeline, leading to a potential difference in the insulating layer. In this paper, based on the ATP-EMTP simulation results, the mine current of $T_0$ and $T_{R1}$ tower grounding grid is introduced into the COMSOL model.

When the lightning impulse current disperses through the tower grounding grid, it will cause a nearby rise in ground potential. The equation for calculating the potential $V_t$ is shown in Equation (1):

$$V_t = k \times i_t \times R \tag{1}$$

where $k$ is the impact coefficient, $i_t$ is the current into the mine through the tower, and $R$ is the tower grounding resistance.

Without considering the tower foundation, the grounding resistance $R$ of the power tower can be calculated according to term Equation (2) when the grounding device is a box extension shape.

$$R = \frac{\rho}{2\pi L}\left(\ln\frac{L^2}{hd} + A_t\right) \tag{2}$$

In term Equation (2), $L = 4(l_{ga} + l_{g1})$, where $l_{ga}$ and $l_{g1}$ are the length of the frame and the epitaxial ray, respectively; h is the buried depth of grounding device; $d$ is the diameter of the grounding body; and $A_t = 1.76$.

The ground potential Vt near the grounding grid will increase due to the dispersion of lightning current. However, as the distance from the grounding grid increases, the ground potential will gradually decrease. The equation for calculating the ground potential rise caused by resistive coupling at a point outside the insulating layer of a gas pipeline is represented by Equation (3).

$$V_x = V_t\frac{r_e}{r_e + x} \tag{3}$$

In Equation (3), $x$ is the distance from the point to the tower grounding device and $r_e$ is the radius of the equivalent hemispherical grounding body of the tower. Its $r_e$ calculation term equation is shown in term Equation (4).

$$r_e = \frac{L}{\left(\ln\frac{L^2}{hd} + A_t\right)} \tag{4}$$

The theoretical term equation suggests that the lightning overvoltage of gas pipelines is influenced by factors such as the spacing between pipelines, the amplitude of lightning current, and soil parameters. Therefore, a simulation analysis will be conducted to examine these influencing factors.



### 4.1. The Influence of Lightning Peak Current

In order to investigate the impact of lightning peak current on lightning overvoltage of natural gas pipelines at the cross-crossing under the influence of lightning stroke conductor, certain parameters were fixed: soil resistivity (ρ) of 1000 Ω·m, spacing between pipelines ($d_{gp1}$) of 40 m, and cross angle (β) of 0°. Lightning peak currents of 25 kA, 30 kA, 35 kA, 40 kA, 45 kA, and 50 kA were sequentially selected. The lightning current waveforms of $T_0$ and $T_{R1}$ tower grounding grids under different lightning peak currents were imported into COMSOL Multiphysics. Figure 5 illustrates these waveforms. The lightning overvoltage amplitude of natural gas pipelines at the cross-crossing of overhead lines is depicted in Figure 7. Figure 8 presents the potential height distribution of the tower grounding grid and crossing natural gas pipeline under different lightning peak currents.

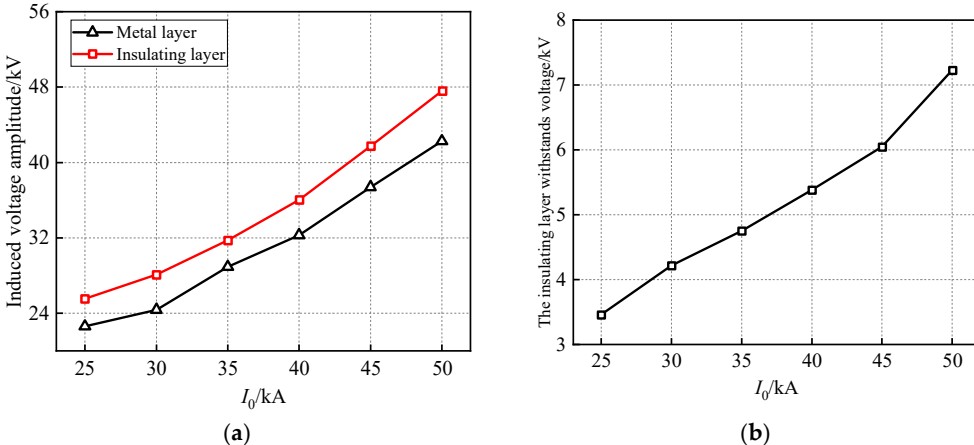

(**a**)  (**b**)

**Figure 7.** The influence of lightning peak current on the overvoltage of natural gas pipelines at cross-crossings. (**a**) Induced voltage amplitude of metal layer and insulation layer of pipeline. (**b**) Voltage amplitude of pipeline insulation layer.

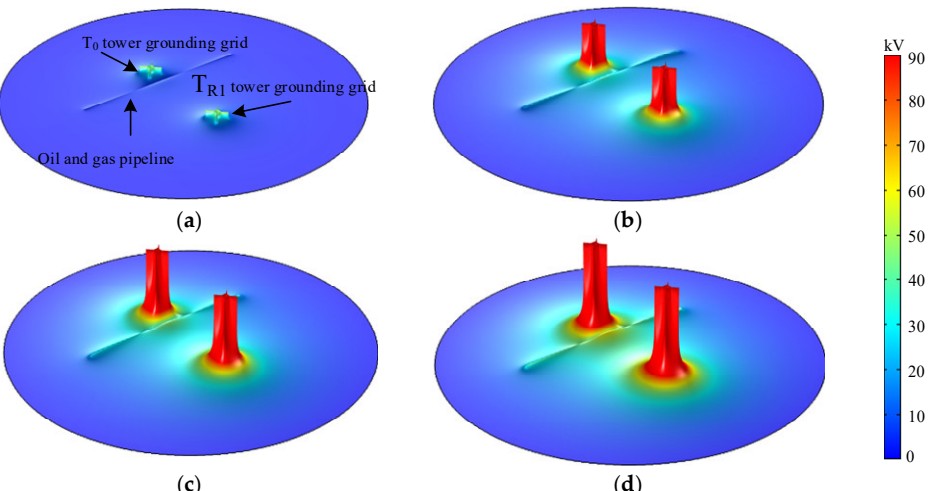

(**c**)  (**d**)

**Figure 8.** The potential height distribution of tower grounding grid and natural gas pipeline at cross-crossing under different lightning amplitudes. (**a**) I0 = 25 kA. (**b**) I0 = 30 kA. (**c**) I0 = 40 kA. (**d**) I0 = 50 kA.

Based on the results shown in Figure 7a, it can be observed that as the lightning peak current increases, both the induced voltage amplitude of the metal layer and the induced voltage amplitude of the insulation layer of the natural gas pipeline also increase. Specifically, when the lightning peak current is raised from 25 kA to 50 kA, the induced voltage amplitude of the metal layer rises from 22.61 kV to 42.25 kV, while the induced

voltage amplitude of the insulation layer increases from 25.55 kV to 47.56 kV. Similarly, Figure 7b illustrates a similar trend in the voltage amplitude of the pipeline insulation layer with respect to the lightning peak current. In this case, as the lightning peak current increases from 25 kA to 50 kA, the voltage amplitude of the pipeline insulation layer increases from 3.46 kV to 7.23 kV, representing a significant increase of 108.96%.

The reason for studying the variation law mentioned above is that as the amplitude of the lightning current increases, the current flowing into the earth through the tower also increases. This, in turn, leads to an increase in the potential of the tower grounding grid and the surrounding soil, under the same soil resistivity. Consequently, the resistive coupling effect of the gas pipeline is aggravated. Equation (1) demonstrates that this rise in potential will cause the nearby ground potential to increase, resulting in a larger potential difference between the outer insulation layer and the metal layer of the pipeline. Thus, it is evident that the amplitude of the lightning current directly influences the voltage amplitude of the pipeline insulation layer.

Based on the results shown in Figure 8, it is evident that the potential height of the $T_0$, $T_{R1}$ tower grounding grid and the surrounding soil increases significantly with the increase in lightning peak current. This can be attributed to the increased flow of scattered lightning through the tower grounding grid. When the soil resistivity remains constant, the potential of both the tower grounding grid and the surrounding soil increases, thereby exacerbating the resistive coupling effect on the natural gas pipeline. At the same time, it can be obviously seen that the potential height at the center of the natural gas pipeline is significantly higher than the potential height at both ends of the natural gas pipeline.

### 4.2. The Influence of Soil Resistivity

In order to investigate the impact of soil resistivity on the overvoltage of natural gas pipelines at the cross-crossing during lightning strikes, several parameters were set as follows: the lightning peak current ($I_0$) was fixed at 50 kA, the distance between the "tube-line" ($d_{gp1}$) was set to 40 m, and the cross angle of the "tube-line" ($\beta$) was 0°. The soil resistivity ($\rho$) was varied, with values of 200 Ω·m, 500 Ω·m, 800 Ω·m, 1000 Ω·m, 1200 Ω·m, and 1500 Ω·m being selected sequentially. The current waveform of the lightning current passing through the $T_0$ and $T_{R1}$ tower grounding grid under different soil resistivity conditions is shown in Figure 6. The resulting lightning overvoltage amplitude of the natural gas pipeline at the cross-crossing is presented in Figure 9. Additionally, Figure 10 illustrates the potential distribution of the tower grounding grid and the natural gas pipeline at the cross-crossing for different soil resistivity values.

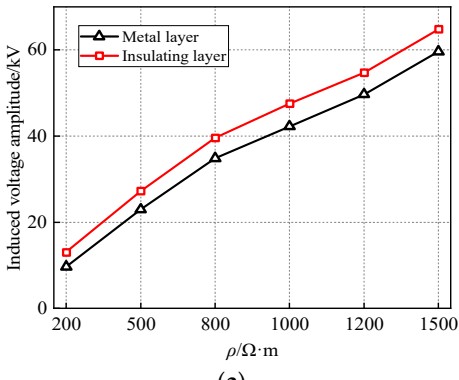

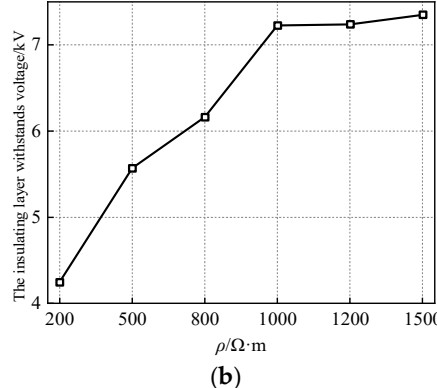

(a)  (b)

**Figure 9.** The influence of soil resistivity on overvoltage of natural gas pipeline at cross-crossing. (**a**) Induced voltage amplitude of metal layer and insulation layer of pipeline. (**b**) Voltage amplitude of pipeline insulation layer.

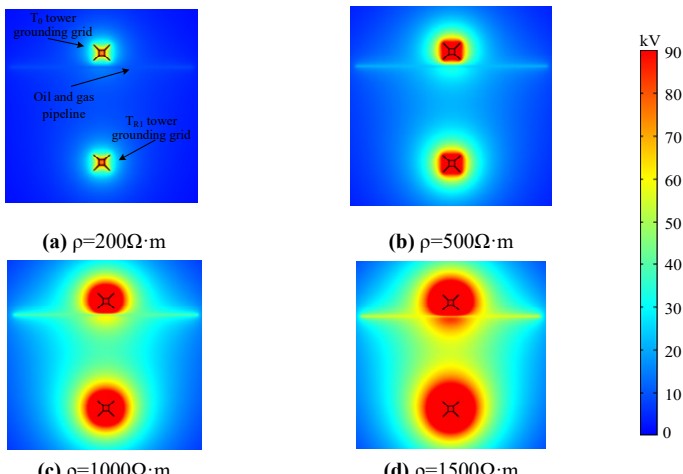

**(a)** ρ=200Ω·m  **(b)** ρ=500Ω·m

**(c)** ρ=1000Ω·m  **(d)** ρ=1500Ω·m

**Figure 10.** Potential distribution of tower grounding grid and natural gas pipeline at cross-crossing under different soil resistivity.

According to Figure 9a, it can be observed that the induced voltage of the metal layer and insulation layer of the natural gas pipeline increases as the soil resistivity increases. When the soil resistivity increases from 200 Ω·m to 1500 Ω·m, the induced electric potential amplitude of the natural gas pipeline and the induced voltage amplitude of the insulation layer increase from 9.68 kV and 13.07 kV to 59.63 kV and 64.80 kV, respectively. This represents an increase of 516.01% and 395.79% for the induced electric potential amplitude and induced voltage amplitude, respectively. The results in Figure 9b demonstrate that as soil resistivity increases, the pressure resistance of the pipeline insulation layer gradually increases. However, after reaching a certain point, the increase in pressure resistance becomes less pronounced. For instance, when the soil resistivity increases from 200 Ω·m to 1000 Ω·m, the withstand voltage of the insulating layer increases from 4.25 kV to 7.23 kV, representing a 70.12% increase. On the other hand, when the soil resistivity increases from 1000 Ω·m to 1500 Ω·m, the withstand voltage of the insulating layer only increases from 7.23 kV to 7.35 kV, resulting in a mere 1.66% increase.

The reason for studying the variation law mentioned above is that an increase in soil resistivity can impede the spreading of the tower grounding grid to the surrounding soil. As a result, the mine current is more likely to spread towards the far end of the extension ray, leading to an increase in the effective spreading area of the tower grounding grid and the soil potential near the grounding grid. This, in turn, exacerbates the impact of the tower grounding grid on the gas pipeline. When the soil resistivity prevents the tower grounding grid from dispersing into the surrounding soil, the induced voltage and withstand voltage of the metal layer and insulating layer of the gas pipeline increase. Conversely, when the mine current decreases, the induced voltage and withstand voltage of the metal layer and insulating layer of the gas pipeline decrease. Therefore, the interaction of these two factors causes the induced voltage and withstand voltage of the metal layer and insulating layer of the gas pipeline to exhibit an upward trend.

Figure 10 illustrates that an increase in soil resistivity obstructs the spread of the tower grounding grid box into the surrounding soil. Consequently, the mine current tends to spread towards the far end of the extension ray such that the effective spreading area of the tower grounding grid and the soil potential near the grounding grid will increase, which will lead to the aggravation of the influence of the tower grounding grid on the natural gas pipeline.

### 4.3. The Influence of "Tube-Line" Spacing

The study investigates the impact of different "tube" spacings ($d_{gp1}$) on the overvoltage of a natural gas pipeline when subjected to lightning strikes. The fixed soil resistivity (ρ) is 1000 Ω·m and the lightning peak current ($I_0$) is 50 kA. The "tube" cross angle (β) is 0°. The

"tube" spacing options considered are 40 m, 50 m, 60 m, 70 m, 80 m, and 90 m. Figure 11 illustrates the amplitude of lightning-induced overvoltage on the natural gas pipeline at the intersection. Figure 12 shows the distribution of potential height for the tower grounding grid and the intersecting natural gas pipeline under different "pipeline" spacings.

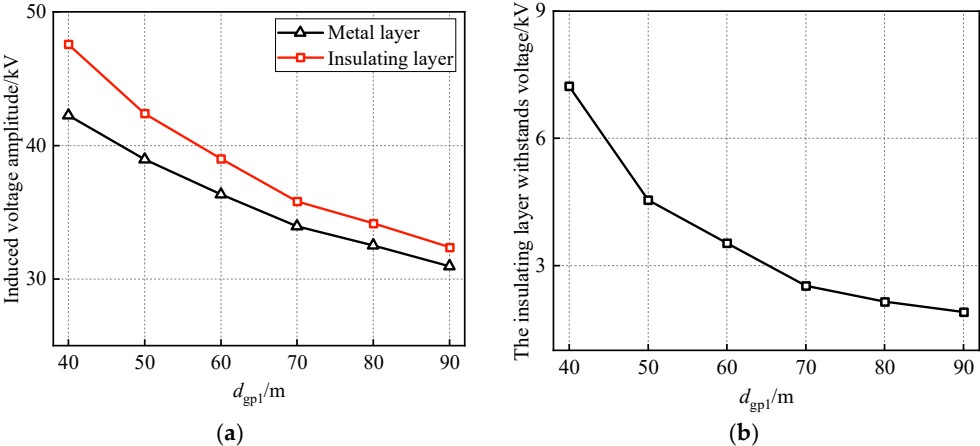

(a) (b)

**Figure 11.** Influence of "pipeline" spacing on overvoltage of natural gas pipeline at cross-crossing. (**a**) Induced voltage amplitude of metal layer and insulation layer of pipeline. (**b**) Voltage amplitude of pipeline insulation layer.

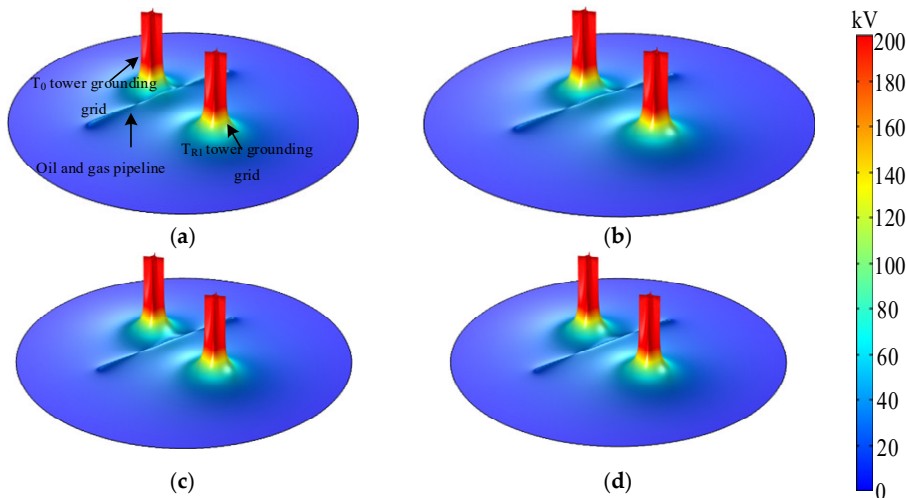

**Figure 12.** The potential height distribution of tower grounding grid and natural gas pipeline at cross-crossing under different "pipeline" spacings. (**a**) $d_{gp1}$ = 40 m. (**b**) $d_{gp1}$ = 50 m. (**c**) $d_{gp1}$ = 70 m. (**d**) $d_{gp1}$ = 80 m.

Based on the findings of Figure 11a, it was observed that as the spacing between pipelines increases, the induced electric potential of both the metal layer and the insulation layer of the natural gas pipeline decreases. Specifically, when the pipeline spacing increases from 40 m to 90 m, the induced voltage of the metal layer decreases from 42.25 kV to 30.96 kV and the induced voltage of the insulating layer decreases from 47.56 kV to 32.37 kV; these reductions correspond to a decrease of 26.72% and 31.94%, respectively. Figure 11b illustrates that the withstand electric potential of the pipeline insulation layer gradually decreases with the increase in pipeline spacing. However, as the spacing continues to increase, the voltage drop trend of the insulation layer becomes more gradual. For instance, when the spacing increases from 40 m to 70 m, the withstand voltage of the insulating layer decreases from 7.23 kV to 2.53 kV, a decrease of 65.01%. When the spacing further increases from 70 m to 90 m, the withstand voltage of the insulating layer decreases from 2.53 kV to 1.91 kV, representing a decrease of only 24.51%.

The reason for studying the variation law mentioned above is that the pipeline's center is closer to the tower grounding grid and the effective dispersion area of the tower grounding grid. This proximity causes a more significant impact of the tower grounding grid dispersion on the center of the pipeline. When the spacing between the pipeline and the tower is small, the center of the pipeline is more affected by the dispersion of the tower grounding grid at both ends. As a result, the induced voltage curve along the insulation layer becomes more prominent. However, when the spacing between the pipeline and the tower increases within a certain range, the difference in the impact of the tower grounding grid dispersion on the center of the pipeline and the two ends reduces. Consequently, the overall curve of the induced voltage along the insulation layer becomes relatively flat.

Figure 12 demonstrates that altering the "tube-line" spacing dgp1 does not impact the potential height distribution of the grounding grid and the surrounding soil area. In order to ensure that the lightning current components of the scattered current through the $T_0$ and $T_{R1}$ tower grounding grids are approximately equal, the vertical distance between the natural gas pipeline and the $T_0$ and $T_{R1}$ tower grounding grids can be adjusted by changing the "tube-line" spacing $d_{gp1}$. This adjustment will affect the extent to which the scattered current from the $T_0$ and $T_{R1}$ grounding grids impacts the natural gas pipeline. Consequently, a minimum value of pipeline overvoltage can be achieved in this scenario.

## 5. Pipeline Overvoltage Protection Based on Solid-State Decoupler

The solid-state decoupler is a product designed to protect against AC/DC interference. It combines embedded current drainage, surge protection, and AC decoupling, and has a high capacity for breaking impact currents. Internally, it typically consists of capacitors, thyristors (or diodes), and surge protection devices connected in parallel. Figure 13 illustrates the appearance and internal circuitry of the solid-state decoupler, while Table 3 provides its technical parameters.

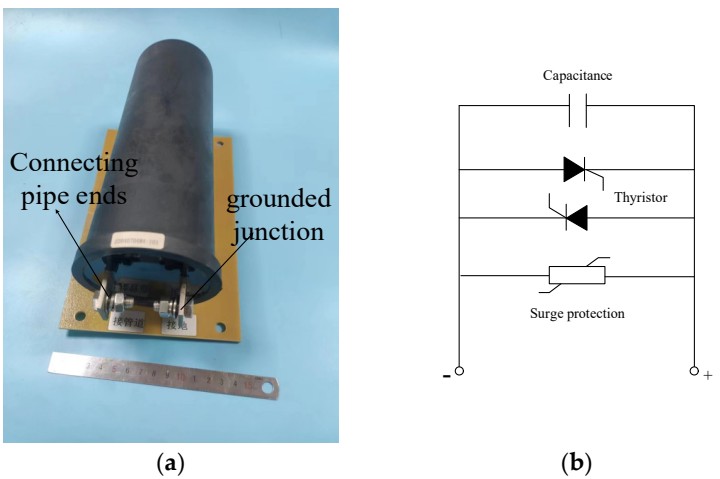

(**a**)　　　　　　　　　　　　　　　　(**b**)

**Figure 13.** Solid-state decoupler. (**a**) Physical images. (**b**) Circuit diagram in thyristor mode.

**Table 3.** Related technical parameters of solid-state decoupler.

| DC Leakage/mA | ≤1mA | Isolation Voltage/V | +2/−2 or +1/−3, etc. | Impact Flow Capacity/ (8/20 μs)/kA | 100 |
|---|---|---|---|---|---|
| Steady-state alternating current/A | 45 | Working temperature/°C | −45~60 | Protection level | ≥IP65 |
| Fault current (AC-rms/power frequency/30 cycle wave)/A | | | | | ≥3500 |

This section aims to decrease the electric potential of the pipeline insulation layer by utilizing the specific wiring method of the solid-state decoupler. It also seeks to validate

the solid-state decoupler's ability to reduce the voltage of the pipeline insulation layer through modeling and simulation. Figure 14a illustrates the simulation diagram. In this setup, a bare copper wire is placed parallel to the natural gas pipeline, with a distance of 1m between them. The length of the bare copper wire is determined based on the current dispersion characteristics of the tower grounding grid. In order to save cost and avoid excessive design, the laying length of bare copper wire $pl_1$ is defined as shown in Figure 14b. The calculation term equation of laying length of bare copper wire is shown in term Equation (1). In this section, $pl_1 = 0$ m indicates that there are no protective measures for natural gas pipelines.

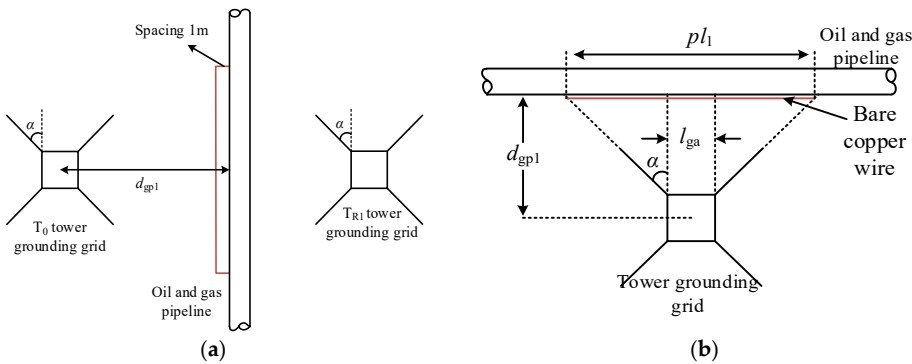

**Figure 14.** Solid-state decoupler wiring diagram. (**a**) Simulation diagram. (**b**) Bare copper wire length diagram.

$$pl_1 = 2 \times (d_{gp1} - 0.5 \times l_{ga}) + l_{ga} \tag{5}$$

In term Equation (5), $pl_1$ is the length of bare copper wire laying, $d_{gp1}$ is the "tube-line" spacing, and $l_{ga}$ is the length of the epitaxial ray.

To evaluate the effectiveness of the method, a simulation model is created using COMSOL Multiphysics simulation software, as illustrated in Figure 14a. The aim is to analyze the impact of varying laying lengths of bare copper wire ($pl_1$) on the resistive coupling overvoltage of natural gas pipelines at the cross-crossing. The simulation parameters used are as follows: lightning peak current ($I_0$) = 50 kA, soil resistivity ($\rho$) = 1000 $\Omega \cdot$m, spacing between the tube and the line ($d_{gp1}$) = 40 m, length of the epitaxial ray ($l_{g1} = l_{g2} = 20$ m), commutation angle ($\alpha$) = 45°, and cross angle between the tube and the line ($\beta$) = 0°. The laying lengths of bare copper wire considered are $pl_1 = 0$ m, 30 m, 40 m, 50 m, 80 m, and 200 m. Figure 15 depicts the overvoltage amplitude of natural gas pipelines under different lengths of bare copper wire, as determined through simulation calculations.

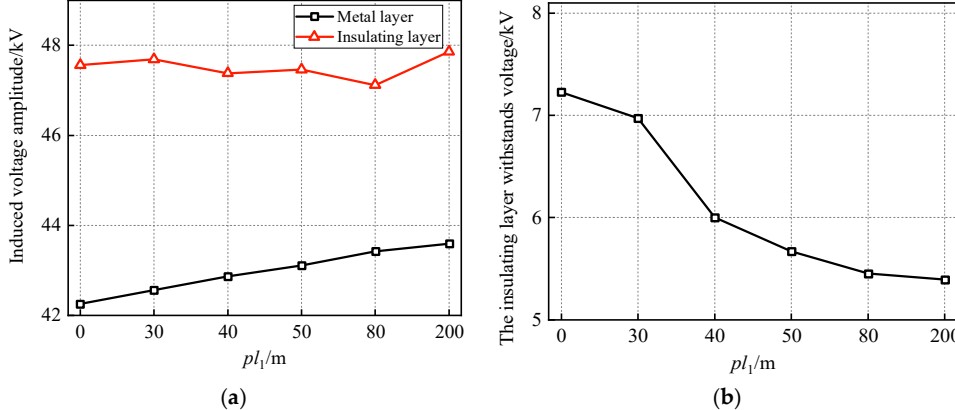

**Figure 15.** Influence of bare copper wire length on overvoltage of natural gas pipeline. (**a**) Induced voltage amplitude of metal layer and insulation layer of pipeline. (**b**) Voltage amplitude of pipeline insulation layer.

According to the results shown in Figure 15, as the length of the bare copper wire increases, the induced voltage of the metal layer of the natural gas pipeline also increases gradually, while the voltage of the pipeline insulation layer decreases gradually. This phenomenon occurs because the solid-state decoupler causes an instantaneous short-circuit between the bare copper wire and the pipeline when subjected to lightning impulse current. The soil potential in the area where the bare copper wire is located is much higher than the induced voltage of the pipeline metal layer, leading to an increase in the induced voltage of the pipeline metal layer during the short-circuiting process. On the other hand, the induced electric potential of the pipeline insulation layer remains relatively stable and fluctuates within a certain range, which can be disregarded. The withstand electric potential of the insulation layer depends on the difference between the induced voltage of the pipeline insulation layer and the metal layer. Therefore, when the induced voltage of the insulation layer changes slightly and the induced electric potential of the metal layer continues to increase, the withstand voltage of the insulation layer gradually decreases.

When the bare copper wire length ($pl_1$) is 0 m, the induced voltage amplitude of the metal layer of the natural gas pipeline is 42.25 kV and the voltage amplitude of the insulating layer is 7.23 kV. When the bare copper wire length is 80 m, the induced voltage amplitude of the metal layer of the natural gas pipeline is 43.43 kV and the voltage amplitude of the insulating layer is 5.45 kV. Similarly, when the bare copper wire length is 200 m, the induced voltage amplitude of the metal layer of the natural gas pipeline is 43.59 kV and the voltage amplitude of the insulating layer is 5.39 kV. It can be observed that increasing the length of the bare copper wire from 0 m to 80 m results in a decrease in the withstand voltage of the pipeline insulation layer from 7.23 kV to 5.45 kV, representing a decrease of 24.62%. However, when the length increases from 80 m to 200 m, the withstand voltage of the pipeline insulation layer decreases only slightly from 5.45 kV to 5.39 kV, representing a decrease of 1.10%.

To better understand the impact of the external bare copper wire of the solid-state decoupling device on the overvoltage of the natural gas pipeline, in close proximity, we focused our research on the central section of the tower grounding grid. This section, which is perpendicular to the natural gas pipeline, was chosen as the research path. Figure 16 illustrates this selection, with a research path length ($l_r$) of 400 m and depth ($h_3$) = 2 m. By conducting simulation calculations, we obtained the soil potential change curve for the path under different lengths of the bare copper wire ($pl_1$), as depicted in Figure 17.

Figure 17 demonstrates that the implementation of solid-state decoupling primarily alters the soil potential surrounding the bare copper wire and the pipeline location. However, it has minimal impact on the soil potential of the tower grounding grid and other regions. As the length of the bare copper wire increases, the soil potential around it gradually decreases. Conversely, the soil potential at the pipeline position exhibits a gradual increase.

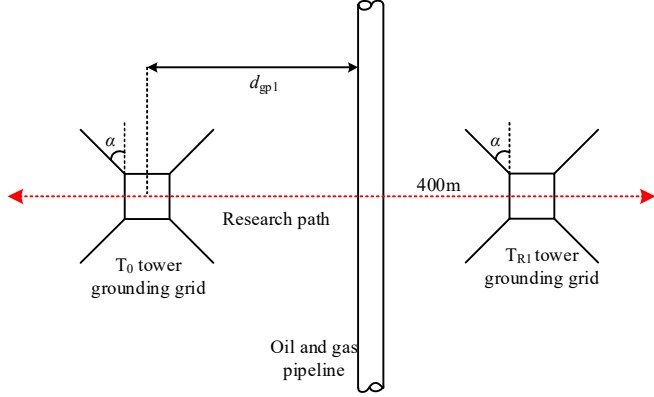

**Figure 16.** Research path top view.

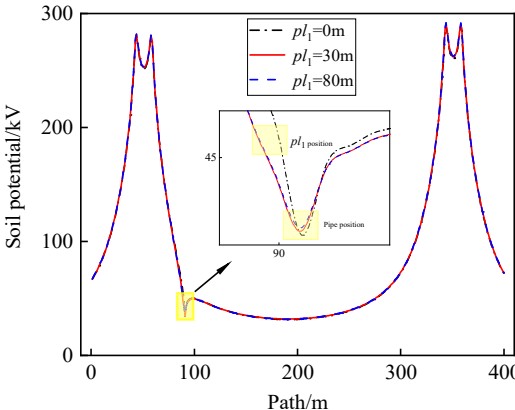

**Figure 17.** The influence curve of bare copper wire length on soil potential in the research path.

The subsequent increase of 120 m of bare copper wire has minimal impact on reducing the voltage of the pipeline insulation layer. Therefore, increasing the length of the bare copper wire does not improve the overvoltage protection effect of the pipeline. In practical engineering, it is more economical and convenient to determine the length of the bare copper wire according to Figure 14b. The detailed calculation term equation is shown in term Equation (5).

## 6. Conclusions

The current research on overvoltage of adjacent pipelines caused by lightning strikes on overhead lines only focuses on the situation of lightning strikes on towers. When a lightning strike occurs on the conductor and causes the insulator to flashover, there is grounding scattered current on both sides of the lightning strike point. This scattered lightning current can also generate pipeline overvoltage on the natural gas pipeline at the intersection. Additionally, to prevent excessive design of pipeline overvoltage protection measures, this paper presents the optimal length term equation of bare copper wire for a solid-state decoupling device. The conclusions of this study are as follows:

(1)  As the lightning peak current increases, there is an upward trend observed in the induced voltage of the metal layer, the induced voltage of the insulation layer, and the withstand voltage of the insulation layer in natural gas pipelines. Specifically, when the lightning peak current increases from 25 kA to 50 kA, the withstand electric potential of the insulation layer increases by 108.96%.

(2)  The induced voltage of the metal layer, the induced voltage of the insulating layer, and the withstand electric potential of the insulating layer of the natural gas pipeline exhibit an upward trend with increasing soil resistivity. As the soil resistivity increases from 200 $\Omega$ m to 1500 $\Omega$ m, the electric potential of the insulating layer increases by 72.94%.

(3)  As the spacing between the "pipeline" increases, the induced voltage of the metal layer, the induced electric potential of the insulating layer, and the withstand electric potential of the insulating layer of the natural gas pipeline gradually decrease.

(4)  In situations where there is scattered lightning current in the ground, the voltage of the pipeline's insulation layer can be reduced by using a solid-state decoupling device, a bare copper wire, and a specific connection mode. When the length of the bare copper wire is increased from 0 m to 80 m, the voltage amplitude of the pipeline's insulation layer decreases by 1.78 kV.

**Author Contributions:** Conceptualization, W.L. and Z.J.; methodology, Y.H. and H.T.; software, H.T.; validation, H.T. and W.L.; formal analysis, Y.H.; investigation, X.S. and Y.H.; resources, X.S.; data curation, Y.H.; writing—original draft preparation, H.T.; writing—review and editing, Y.H.; visualization and supervision, J.X. and W.S.; project administration, Y.W.; funding acquisition, Z.J. All authors have read and agreed to the published version of the manuscript.

**Funding:** This project is supported by the State Grid Hubei Electric Power Co., Ltd. Science and technology project. grant number (B315F0222836).

**Institutional Review Board Statement:** Not applicable.

**Informed Consent Statement:** Not applicable.

**Data Availability Statement:** The data presented in this study are available in article.

**Acknowledgments:** The authors would like to express sincere thanks to the Science and Technology Research Institute, National Oil and Natural Gas Pipeline Network Group Company Limited, and CNPC Engineering Materials Research Institute Co., Ltd., State Key Laboratory of Service Behavior and Structural Safety of Petroleum Pipe and Equipment Materials.

**Conflicts of Interest:** The authors declare that they have no known competing financial interests or personal relationships that could have appeared to influence the work reported in this paper. Wei Liu, Zhipeng Jiang, Xiaole Su, Jie Xiong, Wei Su and Yi Wang are employee of State Grid Hubei Power Supply Limited Company Ezhou Power Supply Company, who provided funding and teachnical support for the work. The funder had no role in the design of the study; in the collection, analysis, or interpretation of data, in the writing of the manuscript, or in the decision to publish the results.

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
