# Peer review of "Research on Lightning Overvoltage Protection of Line-Adjacent Pipelines Based on Solid-State Decoupling"

_applsci, doi:10.3390/app132212529_

Round 1
Reviewer 1 Report
Comments and Suggestions for Authors
The language is not up to the mark
Even the starting lines of abstract are incorrect
Rather than formula, one should use the term equation
And the variables of equation need to be defined as well, in addition to the reference from where the equation is extracted
Technically, the topic is interesting
Validation of results is missing
The authors have presented their result but where is the analysis and validation
A strong mathematical or logical reasoning is missing that supports the results of simulation
Comments on the Quality of English Language
Needs improvement
Reviewer 2 Report
Comments and Suggestions for Authors
The paper presented an analysis about building the calculation model of the grounding grid-natural gas pipeline and the influence of lightning peak current, soil resistivity, and 'pipe-line' spacing on pipeline overvoltage. This topic is very useful for the operation of the long natural gas pipe line placed adjacent to transmission lines. However, the presentation quality of the paper is very low. I cannot see any references to the main topic of the paper. The paper is considered a technical report on testing the high voltage for the pipeline insulation layer.
Comments on the Quality of English LanguageThe Quality of English is very low.
Reviewer 3 Report
Comments and Suggestions for Authors
The paper's subject is interesting and points out a necessary problem for protecting natural gas pipelines.
The reviewer thinks that the authors should answer some questions to increase the scientific value of the study.
- “The results of this study are summarized in the following table” is written in line 64. Which table? There is no table below this line.
- “proposes three methods for the protection of the three methods at the insulated joints of the pipe.” is written in line 70. What does this sentence mean?
- “In this paper, the 500kV overhead line is selected” is written in line 80. However, the authors didn’t give any reason why they chose this voltage level. The Authors should explain why this voltage level was chosen.
- “The tower is chosen to be more realistic equidistributed multi-wave impedance tower model” is written in line 94. The “more realistic” description doesn’t mean scientific research. Authors should explain this selection reason with detailed information or a mathematical model.
- “As shown in Figure 2, The length of grounding grid box, extended ray and extended ray is 15 m and 20 m respectively.” is written in line 102. But in Figure 2 grounding grid box and extended ray are not shown.
- There is no common unity of expressions for the usage of units in the study. In line 127 “overhead line is 24.18kA” and in line 128 “less than 24.18 k A” expressions were used.
- In Figure 5 “distribution of T0 and TR1 towers” is written in line 134. After 4 lines below “overhead wire span between T0 and TR1 towers” is written in line 138. In the study, these subscript problems are very common.
Comments on the Quality of English LanguageEnglish is difficult to understand in the manuscript and needs extensive editing of language.
Round 2
Reviewer 2 Report
Comments and Suggestions for Authors
The authors incorporated the reviewer's comments into the revised manuscript. Thank you for your efforts!
Comments on the Quality of English LanguageThe English should be revised again to improve the quality of the manuscript.
Reviewer 3 Report
Comments and Suggestions for Authors
The reviewer thanks the authors for making significant changes to the work after the revision.
However, there are still serious errors in the study that should not be included in a scientific article.
- In the Abstract section, the authors used expressions with the subject "we" in the newly added section, which should not be in a scientific article. “In this study, we conduct simulation calculations to analyze the overvoltage experienced by the pipeline 12 due to the dispersion of grounding current from the tower. Additionally, we propose a method for protecting the pipeline from such overvoltage.
- Although the authors were warned that they wrote the units incorrectly after the numerical values in the study, it is seen that there are still the same incorrect spellings in many parts of the study. “When the lightning peak current increases from 25kA to 50kA, the induced voltage amplitude of the metal layer and the induced voltage amplitude of the insulation layer increase from 22.61kV and 25.55kV to 42.25kV and 47.56kV, respectively.”
Comments on the Quality of English LanguageEnglish is still too difficult to understand in the manuscript and needs extensive editing of language.
